# OpenReview forum: "Learning with convolution and pooling operations in kernel methods"
_ICLR.cc/2022/Conference — ICLR 2022 Submitted_

### Official Review · Reviewer_HeDj · 2021-10-29

**Correctness:** 3
**Technical Novelty And Significance:** 3
**Empirical Novelty And Significance:** 2
**Recommendation:** 5
**Confidence:** 4

**Main Review:**

To be precise:

- I’m a bit curious about a couple of statements of this paper, as for instance pooling/down-sampling to be an important component of CNNs. I understand that for speed purposes, traditional CNNs tend to reduce the resolution of their signal progressively. However, in general, pooling or downsampling are not desirable because they imply a relative loss of information. If the motivation to use those operators is really to “bias the signal toward low-frequencies”, why don’t we apply a low-pass filtering before feeding a RKHS model with signals like images? I think the reason is obvious, because the loss of information is too important. While I could also understand why this could be(and not “requires to be”) progressively incorporated, this paper completely ignores this aspect and simply advocates that pooling/downsampling are important (without saying precisely why)
As a suggestion, one way to deal with the loss of info is to consider the lost-frequencies and to propagate them in the network (for instance, if P is the pooling, I-P could allow to propagate them). Also, note that downsampling can be combined with a high-pass filter (as with conjugate mirror filters, for a wavelet transform) - it seems not obvious to me that this paper is aware of this.

- I think there is a form of paradox in the aforementioned references of the paper, which implies depth and laziness(ie linearization in a neighbourhood of the initialization) are important. First of all, kernel methods which are make “deep” are not restricted to the study of random kernels, see for instance https://arxiv.org/pdf/2003.02237.pdf - Second, (Thiry et al, 2021) seems to be almost identical to (Coates et al, 2011), while obtaining results in the range of (Mairal, 2014; Shankar 2020), without being deep or restricting their model to a specific subspace (beyond learning a linear model). On the contrary, Lazy models (Chizat, 2019), even hierarchical, reach barely the accuracy of non-learned models. Thus, in my view, the simplest instance of CKNs is obtained via (Thiry et al, 2021) (accuracy above 87%), yet I’m not sure how this work brings any new intuitions or explains it. In other words, would it be possible that something simpler than deep CKNs fitted in their RKHS space would also get excellent accuracy while being explained only up to a limited extent by this framework? Would this framework shed some lights on those ideas?

- I’d be surprised that such linear structure is actually present in the data (e.g., the operators to exhibit a fast decay) - this should be numerically demonstrated in order to validate the approach. As this is a motivation of this work, I'd really appreciate to see that actual CNNs/lazy CKNs exhibit this type of behavior. I however understand that taking the data from Unif(Q^d) is a necessary simplifications to write a couple of equations.

[postrebuttal] My opinion, after reading others reviews, is that some claims of the paper are not well motivated or supported because the results are too specific. Also, I unfortunately was only partially convincing by authors' answers.

**Summary Of The Paper:**

This work proposes to study CKNs (defined via random features) which are defined via convolutions on “patches” (e.g., operators that writes (x,y)->\sum_k h(<x_k,y_k>) where $x_k,y_k$ are some patches of $x,y$), and to exploit the fact that those types of kernels are actually related to convolutions on this space, and that one can decompose them on an orthogonal basis of “zonal-like” Polynomials, h(<x,y>)=\sum_n P_n(<x,y>) where $\{P_n\}$ is an orthogonal basis s.t. P_n(<x,y>)=\sum_k Y_k(x)Y_k(y) are some “harmonics-like” functions. Then, one diagonalizes every kernel in this common basis, and all the next bounds exploit this linear structure and a fast decay of the eigen-values, which allows to derive Generalization errors based on this linear structure with a low-dimension. Overall, this is a standard framework (Bach, 2021, 2017) which has been well explained and cited across the text. The link with lazy training is done and the description of the space of L^2 patches(I prefer to phrase it this way rather than L^2 hypercube, I think it’s makes more clear what the authors try to achieve) is well-done, while very similar to L^2(S^{d-1}).

**Summary Of The Review:**

Overall, I believe the paper is well written, technically correct as I checked most of the proofs or I’ve already seen/dones some sketches of similar ideas. I feel no classification experiment on CIFAR-10(or other dataset) is really needed as this algorithm doesn’t try to derive any optimization algorithms, yet rather try to describe the space of functions associated to this framework. I think however that given that this paper tries to explain deep neural networks, it'd be nice to validate this approach empirically. My further concerns are rather linked to the motivations of this study and the choice of locality and/or laziness and/or some modules.

---

> ### Author Response · Authors · 2021-11-13
> **Response to reviewer 3**
>
> Thank you for the review and helpful comments.
> We would like to answer some of the remarks (in the same order as in the review):
>
> F) In our paper, we were mainly motivated by understanding the performance of the current state-of-the-art kernel methods (Shankar et al. 2020, Arora et al. 2020, Bietti 2021), which seems to require local pooling (kernels without local pooling are shown to have worse performance in these same papers). While we agree that biasing towards low-frequency might not always be a good idea (e.g., for deep CNN), on these particular image datasets and for shallow CNTK, it does improve the generalization error.
> Also our results show that local pooling biases the learning (easier to learn low-frequency) without erasing information, and does not preclude learning high frequency components.
>
> Pooling and downsampling operations in CNNs are usually motivated by either dimension reduction and/or making feature position invariant. Here we show (bottom of page 32) that doing only downsampling (dimension reduction) does not impact sample complexity (it can however speed up computation as mentioned by the reviewer), while if the target function is low-frequency then adding local average pooling induces a factor $\omega$ improvement in the sample complexity. Our message is therefore more nuanced: using an architecture that is adapted to the target function will lead to a large improvement in statistical performance. On the other hand, better understanding the RKHS of good-performing kernels could potentially provide information on the properties of image function classes.
>
> We will further clarify these points in a revised version of the paper.
>
> G) The kernels in our paper can be defined on their own without referring to CNNs. We introduce CNTK in order to make the correspondence with CNN architectures and give intuition about these types of kernels. But as mentioned by the reviewer, this is not required, e.g., Shankar et al. (2020).
> As we consider a fixed RKHS setting, our work will have limited relevance to explain data-dependent feature extraction procedures such as the one described in Thiry et al. (2021). Our focus in this paper is however different: it is to better understand the role of the different layer operations, which might still be relevant in feature learning settings. Furthermore, it is yet unclear what the limitations of kernel methods are. For example, Bietti (2021) showed that 3-layers CKN with degree-3 polynomial activation on the second and third layers achieved similar results as state-of-the-art kernel methods (90% on CIFAR-10).
>
> H) Our results do not require fast decay of the eigenvalues. In particular, our high-dimensional results require mild assumptions on the eigenvalues of the kernel (the kernel only need to have bounded trace) and we consider an arbitrary target function. For the source and capacity conditions, $\alpha$ and $\beta$ can be taken arbitrary as long as they verify the constraints in (A1) and (A2).
>
>
> We refer to comment C) for what we believe is the main contribution of our paper (see also points A) and B) for further motivations).

---

> > ### Comment · Reviewer_HeDj · 2021-11-22
> > **Thanks**
> >
> > Thank you for your rebuttal. Sorry for my late answer, I'll take it in account for my final assessment.
> >
> > I'm a bit unclear about the last sentence of F): if the inductive bias is good, learning is indeed easier - yet, the current kernel hasn't been tried on natural images, thus it's unclear if provide relevant information on the target function, or am I missing something?
> >
> > I'm not really convinced by the argument in G) - I guess my point of view can be summarized by the last sentence of the abstract: "our results quantify how choosing an architecture adapted to the target function leads to a large improvement in the sample complexity" - this is a bit tautological: yes if the target function has the good structure, then it'll get easier to learn but this is not, to my understanding, a discovery, yet unfortunately the good hypothesis class of the target function remains mainly still unknown.
> >
> > Concerning H), I guess I'm misunderstanding - I thought the solution of KRR would be related (I have in mind theorem 7/8 in appendix C) to an effective threshold, which seems to require a fast decay of the target function. I guess I misunderstand the statement "it does not fit this eigendirection at all"(p31)
> >
> > Thanks again for your answers.

---

> > > ### Author Response · Authors · 2021-11-23
> > > **Response to reviewer 3 (1)**
> > >
> > > Thank you for the further comments!
> > >
> > > I) The kernels considered in the paper are one-layer convolutional kernels (CK) from Mairal et al. (2014), Mairal (2016), which have been tested on several datasets. For example, our kernel covers the one-layer CK tested in Bietti (2021), which achieves 80.9% accuracy on CIFAR-10 with $3\times 3$ patches and weighted-average local pooling layer (which falls under the setting we can study, see Appendix A.3). Note that Coates et al. (2011) achieved accuracy 79.6%, and was the best non-deep learning method before Mairal et al. (2014). While depth is important in simulations, Bietti (2021) shows that adding a second convolutional layer with a degree 3 polynomial kernel achieves 88.2% accuracy on CIFAR-10 (which matches the 88.2% accuracy of the Myrtle10 in Shankar et al. (2020), the current state-of-the-art kernel , see Tables 1 and 2 in Bietti (2021)). This amounts to adding 3-patches interactions to the RKHS, and as we note in section 2.4, the kernel can still be diagonalized in the data on the hypercube setting, albeit with much more complicated diagonalization matrices. In this work, we wanted instead to get a clear understanding of the roles of convolution/pooling/downsampling operations in convolutional kernel architectures, and focused on the simplest case of one layer CK. The last sentence in F) was intended to provide another high level motivation to study kernel methods (in addition to being interesting methods on their own).
> > >
> > >
> > > J) We agree that the last sentence of the abstract is not a `discovery’, as this is the basic premise for architecture and algorithm design in machine learning. However, understanding the inductive bias and generalization performance of a learning method is far from being simple, and as discussed below, the kernel methods considered in the paper are not arbitrary but motivated by recent empirical work. In general, we think that the interaction between architecture, data distribution and generalization is far from being understood, and is an important theoretical research direction. We believe that our setting provides a useful case study for the following reasons:
> > >
> > > 1) Kernel methods are well understood in terms of training, approximation and generalization, which allows us to focus on the interaction between data-distribution and architecture in the performance of the learning method, and give univocal and precise results. The fact that kernel methods and their convolutional architectures are related to neural networks at the beginning of GD and/or in some optimization regime can provide bridges to other learning methods (this is not however the focus of this work).
> > >
> > > 2) Convolutional kernels introduced in Mairal et al. (2014) have achieved remarkable performance in image classification tasks. Furthermore, there was a renewal of interest in kernel methods due to the neural tangent kernel: plenty of works have since then investigated the performance of kernels both theoretically and empirically. Several papers have noted that kernel methods perform surprisingly well (e.g., Arora et al. (2019)) and there is an active line of research trying to understand how far we can push their accuracies on standard datasets.
> > >
> > > 3) The state-of-the-art kernels are convolutional kernels constructed by stacking convolution/pooling/downsampling operations. There is little theoretical work studying the impact of each of these operations on the performance. Here we take the simplest architecture of this type and we focus on understanding their precise role in the one-layer case. Their role will change in deeper architectures (see comments in section 2.4), but some of the one-layer intuition will still apply. Note that (see above) Bietti (2021) demonstrated that adding three-patches interactions to the one-layer CK is sufficient to achieve near state-of-the-art kernel performance on CIFAR-10.
> > >
> > > 4) We consider a simple data-distribution (uniform on the hypercube): our goal is to precisely characterize the interaction between target function and architecture, and quantify performance for any given target function in $L^2$ (which our high dimensional results do). See comments B) and C) in the response to reviewer 1.

---

> > > ### Author Response · Authors · 2021-11-23
> > > **Response to reviewer 3 (2)**
> > >
> > > K) About Theorem 7 and 8: it is a high-dimensional phenomena of kernel ridge regression, which acts as a shrinkage operator (see equation 42 page 29). This is unrelated to the structure of the target function, which is simply assumed to be a function in $L^2$. “KRR will learn perfectly the low-degree polynomial part of the target function and nothing else” means that the test error is equal to $\Vert f_* - \hat f \Vert_{L^2}^2 \approx \Vert P_{>\ell} f_* \Vert_{L^2}$:  $ \hat f$ coincides exactly with the target function on the subspace spanned by polynomials of degree less or equal to $\ell$, while $\hat f$ is approximately $0$ on the subspace spanned by polynomials of degree greater than $\ell$ (hence the test error is approximately $\Vert P_{>\ell} f_* \Vert_{L^2}$, the high-degree part of the target function).
> > > (Note that while these predictions are derived in high dimension, they match well numerical experiments in low/moderate dimension.)
> > >
> > >
> > > Thanks again for your thoughtful review.

---

### Official Review · Reviewer_nYey · 2021-11-02

**Correctness:** 4
**Technical Novelty And Significance:** 2
**Empirical Novelty And Significance:** 2
**Recommendation:** 5
**Confidence:** 3

**Main Review:**

The paper uses Gegenbauer polynomial expansion of the kernel function to prove bounds on the sample complexity. In particular, since the CNTK is a dot product kernel, one can write it in terms of Gegenbauer polynomials and then use the eigendecomposition of such polynomials to prove the results of this paper.
I was wondering if one could prove the same bound by Taylor expansion instead. The benefit of using Taylor expansion is that the assumption that the inputs are on the hypercube would be unnecessary. In particular, a degree-p monomial <x,y>^p for x,y \in R^d can be decomposed as <x^{\otimes p}, y^{\otimes p}> where x^{\otimes p} is self tensor products of x. These vectors are in dimension d^p which is the same dimension as what you would get from Gegenbauer expansion.

**Summary Of The Paper:**

The paper considers the Convolutional Neural Tangent Kernel (CNTK) of depth one L=1 for the inputs that are distributed over the hypercube {-1,+1}^d. Under this setting, the paper presents the eigendecomposition of the kernel function. The paper also investigates the generalization properties of kernel regression using the depth one CNTK and bounds the risk for regression.

**Summary Of The Review:**

I am wondering if one could prove the results of this paper by Taylor expanding the CNTK kernel. The benefit of using Taylor expansion is that the assumption that the inputs are on the hypercube would be unnecessary. In particular, a degree-p monomial <x,y>^p for x,y \in R^d can be decomposed as <x^{\otimes p}, y^{\otimes p}> where x^{\otimes p} is self tensor products of x. These vectors are in dimension d^p which is the same dimension as what you would get from Gegenbauer expansion.

---

> ### Author Response · Authors · 2021-11-13
> **Response to reviewer 2**
>
> Thank you for the review and helpful comments.
> About the suggestion of using a Taylor expansion, we make the following two comments:
>
> D) The goal of this paper is to fully characterize the RKHS of one layer convolutional kernels and derive precise generalization error bounds. To do so, we need to be able to characterize the eigenfunctions and eigenvalues of the kernel integral operator, which are usually hard to get and are not given by the monomials of a Taylor expansion (in general). For example, this is the reason Bietti (2021), Favero et al. (2021), Mei et al. (2021), Bietti et al. (2021) consider non-overlapping patches and uniform distribution on the sphere or cube.
>
> E) More abstract results are possible. Existence of the eigendecomposition of the kernel operator is guaranteed by general spectral theorems and bounds on the RKHS norm can be written implicitly in terms of a minimization problem (see for example Bietti (2021)). The classical generalization upper bound depends on the (non-explicit) RKHS norm of the target function. Similarly, the high-dimensional results are derived from a general abstract framework in Mei et al. (2021a).
> In this work, we decided instead to explicitly write the RKHS norm and get intuition on the kind of regularization induced by convolution, local pooling and downsampling operations.
>
> See comments A), B) and C) about the contributions of this paper.

---

> > ### Comment · Reviewer_nYey · 2021-11-15
> > **Response to authors**
> >
> > I see.
> >
> > I understand that for binary vectors u and v, Eq(1) holds exactly. My question was, what if you lifted the assumption that u and v are binary vectors and let them be in R^p and then use (truncated) Taylor expansion in place of Eq(1). In this case, Eq(1) would hold up to some small exp(-p) error (depending on function h) but the number of eigenfunctions you get would stay the same, i.e. exp(p). This way Eq(4) would hold approximately with the same number of summands.
> >
> > I guess in your analysis it is important that Proposition 1 would hold exactly and somehow you cannot tolerate error.

---

### Official Review · Reviewer_oiTT · 2021-11-04

**Correctness:** 4
**Technical Novelty And Significance:** 3
**Empirical Novelty And Significance:** 2
**Recommendation:** 6
**Confidence:** 3

**Main Review:**

Main strengths:
* Describes the kernel associated with "simplified" CNNs that allows for more precise descriptions of the roles of convolutional, pooling, and subsampling, and thus allows studying how these different operations affect the properties of the network.
* Gives a clear and intuitive characterization of various special cases of the proposed CNN kernel.
* Paper is well written, and places the results well relative to the prior works on the subject.

Main weaknesses:
* While the paper clearly contains novel results, they are more incremental in nature. The techniques used are similar to prior works on NTK, and the derived implications are close to previous results obtained under slightly different settings (as the authors clearly note where applicable).
* Some of the specific implications themselves are not that surprising. For example, in Mhaskar et. al (cited in the paper) where the approximation properties of deep "convolutional" networks is investigated, they also show these structures avoid the curse of dimensionality (in the approximation theory sense) when the target function and hypothesis class share the same structure of "local" functions.

**Summary Of The Paper:**

The paper studies the generalization properties of simple convolutional neural networks with a single convolutional layer, equipped with some non-linearity, followed by averaged pooling, and then a fully-connected linear layer. The paper builds on the idea of the Neural Tangent Kernel, and studies specifically the generalization properties of the kernels corresponding to the above CNN under various architectural choices (window size, pooling size, stride size) and under the assumption of inputs drawn from the uniform distribution on the hypercube $\\{-1,1\\}^d$. The main findings are that a network as above is suitable for learning "local" functions, i.e., functions that are sums of functions over patches. Specifically, the sample complexity depends on the window size $q$ rather than on input dimension $d$, overcoming the curse of dimensionality associated with learning general classes. Moreover, it is shown that when average pooling is introduced, it corresponds to a bias towards lower-frequency functions as manificested by eigendecomposition of the associated kernel.

**Summary Of The Review:**

The paper is clearly written and has novel though a bit incremental results, which builds on previous works and contain results that are similar in nature. I am leaning towards marginally accepting the paper.

---

> ### Author Response · Authors · 2021-11-13
> **Response to Reviewer 1**
>
> Thank you for the review and helpful comments.
> Here are a few further comments about our results:
>
> A) Here, we consider a simple setting to the extent that we fix the data distribution to be uniform on the hypercube and only study the one-layer case. However, we do not simplify further the architecture (which is essentially the same as the one-layer case of Mairal et al. (2014)). This is in contrast with the generalization bounds for CKNs derived in other work: for the most related to ours: 1) non-overlapping patches in Favero et al. (2021), Bietti (2021); 2) only global pooling in Mei et al. (2021), Favero et al. (2021), Bietti (2021); 3) no downsampling.
>
> B) While we agree that the intuitions of some of our results were already displayed in previous work, we find value in working out fully and rigorously a toy setting and clarify the properties of the different layer operations.  For example, 1) breaking the curse of dimensionality was previously shown either in approximation (these models are not known to be tractable), with non-rigorous statistical methods in Favero et al. (2021) or with some simplifications on the RKHS in Scetbon et al. (2020). 2) A precise description of local average pooling, which biases the RKHS towards low-frequency components by reweighing the discrete Fourier components (to the best of our knowledge, mathematical descriptions of the precise role of local pooling are lacking in the literature). 3) The regularization effects of the convolution and pooling layers are independent (they penalize independently the smoothness and the Fourier content). 4) Downsampling does not degrade the low-frequency information (which agrees with the intuition of throwing out fine grained and redundant details).
>
> C) We believe that our main technical contributions are the pointwise generalization errors (Theorems 2, 3, 5 and Corollary 1, summarized in table 1), which quantify for a given target function the gain in sample complexity associated to those different architectures. These pointwise bounds require different techniques than the classical generalization bounds with source and capacity conditions, which only provide minmax bounds up to multiplicative factors, and are not precise enough for the type of separation results that we are interested in. While the pointwise bounds are derived in high-dimension, the predicted rates agree with numerical simulations in moderate/low dimensions (Figures 1 and 6). To the best of our knowledge, this is the first work showing a statistical advantage of these architectures (for example, https://arxiv.org/pdf/2106.05233.pdf is not powerful enough to show such an advantage, and Mei et al. (2021) only considers full size patches and global pooling).

---

### Public Comment · ~Alessandro_Favero1 · 2021-11-13
**Public Comment**

Dear Authors and Reviewers,

We are the authors of the paper "Locality defeats the curse of dimensionality in convolutional teacher-student scenarios", which is appearing in NeurIPS 2021.

The first main result of this paper, i.e. that "the convolution layer breaks the curse of dimensionality by restricting the RKHS to ‘local’ functions", is also the central point of our recent work. This strong overlap between the two papers is currently not acknowledged in the introduction, nor in the main results section.

Although the authors write, after Theorem 1, that this result "was already noticed in [...] Favero et al. (2021) for slightly simplified architectures", we believe that this terminology does not reflect that the aforementioned result is the main focus of our work.

Moreover, we point out that our results hold for both overlapping and non-overlapping patches.

With best regards,

The authors of "Locality defeats the curse of dimensionality in convolutional teacher-student scenarios"

---

> ### Author Response · Authors · 2021-11-14
> **Response to Alessandro Favero's comment**
>
> Dear A. Favero, F. Cagnetta, M. Wyart,
>
> Thank you for your interest in our paper and for pointing out flaws in our presentation! We are indeed aware of your work and we will further acknowledge your contributions and add comments where it applies. More generally, we will add more discussions comparing our paper to previous work and highlight what our contributions are. We will make sure to not claim novelty for insights that are not our own (e.g., that convolutional kernels break the curse of dimensionality). While our work studied similar objects as in Favero et al. (2021), the focus of our work is vastly different. Our goal is to derive mathematically rigorous quantitative bounds that can give separation in generalization power between different architectures.
>
> Furthermore, our Theorem 1 is different than the results in Favero et al. (2021) in two major ways:
>
> 1) We give a non-asymptotic bound with explicit dependency in both $n$ and $d$, which is minimax optimal up to a constant multiplicative factor (this can be showed by modifying the proof in Appendix B.6 of https://arxiv.org/pdf/2106.07148.pdf). In Favero et al. (2021), only the asymptotic rate in $n$ is obtained.
>
> 2) We use a rigorous capacity/source condition bound while Favero et al. (2021) uses statistical physics heuristics and/or a Gaussian universality conjecture. We included this capacity/source approach as we think that this is the standard (and simple) setting for such bounds and would be valuable to the kernel community. However, we believe that our main technical contributions are the pointwise generalization bounds (Theorems 2, 3, 5 and Corollary 1, summarized in table 1).
>
> Hence, we agree that both our Theorem 1 and Favero et al.’s Theorem 4.1 (and Corollary 6.1.1) give sample rates in the generalization error. However, we believe the assumptions, proof techniques and guarantees to be sufficiently different to justify the inclusion of Theorem 1 in our main text. We will however emphasize that such a rate was obtained in Favero et al. (2021).
>
> More generally, while we agree that Favero et al. (2021) has the setting that is the most related to ours (with one layer convolutional kernel), we believe that the insight that locality (rather than translation invariance) breaks the curse of dimensionality was already understood implicitly or explicitly in multiple works. A short list of references: from an approximation point of view, locality is a standard assumption that breaks the curse of dimensionality (Mhaskar et al., (2016)); Du et al. (2019, https://arxiv.org/pdf/1805.07883.pdf) study the sample complexity in a parametric setting to estimate CNNs;  Kohler, Krzyżak and Walter (2020) shows that CNNs break the curse of dimensionality (in generalization) in a parametric setting; Scetbon et al. (2020) shows that convolutional kernels break the curse of dimensionality for a family of distributions and architectures; Mei et al. (2021) shows that pooling do not break the curse of dimensionality when no locality bias is introduced; and a later reference, Walter (2021, https://arxiv.org/pdf/2106.05233.pdf) shows that CNNs with or without pooling break the curse of dimensionality in a parametric setting. Several of those works precisely characterize the sample rates in terms of smoothness/regularity conditions of the target function and CNN architecture.
>
> Thank you for pointing out the overlapping patches case, described in the appendix of Favero et al. (2021), which we missed in a first reading of the paper. We will modify our response to the reviewers to avoid confusions.
>
> We again thank the authors of "Locality defeats the curse of dimensionality in convolutional teacher-student scenarios” for their interest and feedback.
>
> We hope that the above discussion and described modifications will be satisfactory.
>
> Best regards,

---

> > ### Public Comment · ~Alessandro_Favero1 · 2021-11-16
> > **Public Comment**
> >
> > Thanks for considering our comments and for the detailed reply.
> >
> > With the best wishes,
> >
> > A. Favero, F. Cagnetta, M. Wyart

---

### Decision · Program_Chairs · 2022-01-20

**Decision:**

Reject

**Comment:**

The paper analyzes convolutional kernels and their sample complexity as compared to different architectures, and in particular the effect of pooling. The analysis proceeds by characterizing the RKHS in this setting (for a distribution on the cube) and using results by Mei and others to obtain separation between different architectures.
The reviewers appreciated the fact that this is an example worked out in detail, resulting in a clear message about sample complexity gaps between architectures.
However, there was also concerns that some of the conclusions do appear in previous works, so that there is no surprising insight here.
In future versions, the authors are encouraged to more clearly explain the novel aspects of the paper (as well as where the main technical novelties and tools are).